# High Levels of Physical Activity Reduce the Esthetic Durability of Botulinum Toxin Type A: A Controlled Single-Blind Clinical Trial

**DOI:** 10.3390/toxins15070463

**Published:** 2023-07-19

**Authors:** Omar Neves Morhy, Andréa Lisbôa Sisnando, Mariana Barbosa Câmara-Souza, Ana Claudia Carbone, Giancarlo De la Torre Canales

**Affiliations:** 1Ingá University Center, Department of Dentistry, Uningá, Paraná 87035-510, Brazil; omorhy@gmail.com (O.N.M.); andreasisnando@hotmail.com (A.L.S.); mariana_mbcs@hotmail.com (M.B.C.-S.); dra.anacarbone@hotmail.com (A.C.C.); 2Egas Moniz Center for Interdisciplinary Research (CiiEM), Egas Moniz School of Health & Science, 2829-511 Caparica, Portugal; 3Department of Dental Medicine, Karolinska Institutet, Scandinavian Network for Orofacial Neurosciences (SCON), 141-52 Huddinge, Sweden

**Keywords:** botulinum toxin, electromyography, physical activity, wrinkles

## Abstract

The present study aimed to evaluate the influence of physical activity on the durability of the esthetic effect of botulinum toxin type A (BoNT-A). Sixty female patients were allocated to three groups (*n* = 20) according to their physical activity level (PA): Low PA, Moderate PA, and High PA. All groups received a single injection of onabotulinumtoxinA, considering standardized doses in the frontalis (12U), corrugator supercilia (7U, each), and procerus muscles (4U). Outcomes were measured using electromyography (EMG), Merz 5-point scales, and Face-Q scales (perceived age and lines between eyebrows). A follow-up occurred after 30, 60, and 90 days. EMG results showed a significant decrease in muscle activity in the Low-PA group at all follow-ups compared with the other groups (*p* < 0.001). The Merz scale scores showed that the severity of forehead and glabellar lines significantly improved in the Low-PA group throughout this study compared with the other groups (*p* < 0.001). No significant differences between groups were found in the Face-Q scale for perceived age, while the Face-Q scale for lines between eyebrows showed better results for Low-PA (*p* < 0.01) and Moderate-PA (*p* < 0.01) groups compared to the High-PA group at the 30- and 90-day follow-ups. The durability of the esthetic effect of BoNT-A seems to be negatively influenced by the level of physical activity.

## 1. Introduction

Facial aging is a multifactorial, interrelated, 3-dimensional, complex, and dynamic process that involves soft tissue atrophy, bone resorption and remodeling, muscle alterations (hypertrophy), and imbalance between muscle groups [1,2]. It can be aggravated primarily by smoking, sun damage, poor health, and depression [3]. Esthetic consequences of the aging process in the upper face include wrinkles, pronounced forehead lines that can be accompanied by drooping of the eyebrows, and deeper glabellar lines [4].

Injection of neuromodulators such as botulinum toxin type A (BoNT-A) for the treatment of facial lines is the most frequently performed minimally invasive procedure in the United States and Brazil, accounting for 46.4% and 53.7% of the most common non-surgical procedures, respectively [5,6]. The most frequently treated areas with neuromodulators include the forehead and the glabellar complex because the treatment outcomes are excellent, resulting in a more youthful facial appearance, and because of the safety of the procedure [7,8,9]. Although a consensus [10,11] and clinical trials [8,9] have attempted to improve the esthetics effects of BoNT-A by suggesting specific dose ranges for each muscle group and refining the injection techniques, considering the interaction of contraction patterns of the upper face, the durability of these effects is still a matter of discussion.

Depending on the commercial brand [12], the paralyzing-esthetic effect of BoNT-A can last up to 6 months (on average, three to four months) [13]. The waning of the effect is mainly related to the sprouting of new nerve endings, followed by recovery of function at the original nerve terminal and retraction of the sprouts [14,15]. Notwithstanding, several factors have been reported to influence the durability of the BoNT-A effect, which are related to the handling of the drug (reconstitution and storage) [16], the content of the auxiliary protein [17], the amount of active BoNT-A [17], the dosage and intervals between applications [16], and the skin type [18]. In addition, other factors have been pointed out by injectors that are indirectly related to BoNT-A’s esthetic durability in clinical practice, such as smoking, sun exposure, and physical activity level; however, there is no evidence in the current literature of the influence of these factors on BoNT-A durability.

Although the influence of physical activity on the durability of BoNT-A effects has not been fully elucidated, it has been clinically observed that the effects last for less time in people with high levels of physical activity. Furthermore, repetitive motor training in mice (treadmill running or wheel running) has been shown to provide sufficient stimulation of specific neural pathways to facilitate functional reorganization within the spinal cord and improve motor output [19]. Also, clinical studies have shown that individualized exercise programs as adjunctive treatment improve motor recovery in people with spinal cord injuries [20,21]. Although there is no consensus among available studies, plasma and muscle levels of some myokines that promote muscle reinnervation have been shown to increase after exercise, regardless of the type of exercise performed [22,23,24].

Consequently, it can be hypothesized that the effects of high levels of physical activity on muscle reinnervation negatively influence the durability of the esthetic effects of BoNT-A applications. Therefore, the present study aimed to evaluate the influence of physical activity levels on the durability of the esthetic results of BoNT-A applications in the upper face muscles.

## 2. Results

A total of 92 volunteers were screened, and after the inclusion and exclusion criteria assessment, 60 participants (mean age 33.76 ± 6.04 years) were included in this study. Participants were divided into three groups (*n* = 20) according to their physical activity (PA) level: Low PA, Moderate PA, and High PA. No significant differences were found between groups regarding age (*p* > 0.05).

### 2.1. Electromyography Activity (EMG)

Considering each group, a significant decrease in EMG activity was found for the frontalis, corrugator supercilii, and procerus muscles after the treatment in all follow-ups (1, 2, and 3 months) (*p* < 0.0001) (Table 1). For between-group comparisons, no significant differences were found at baseline (*p* > 0.05) for all muscles. However, the Low-PA group presented significantly lower scores for the frontalis and corrugator supercilii muscles at the 1-, 2-, and 3-month follow-up visits (*p* < 0.01) compared to the other groups. Additionally, no significant differences were found at the 2- and 3-month evaluations for the procerus muscle between Low-PA and Moderate-PA (*p* > 0.05) groups, but significantly lower scores were found in the Low-PA group compared with the High-PA group for the same muscle and evaluation periods (*p* < 0.05) (Table 1).

### 2.2. Severity of Forehead and Glabellar Lines

Regarding the intragroup results, the severity of forehead and glabellar lines significantly improved in the Low-PA group throughout the study timepoints compared to baseline (*p* < 0.001). However, in the Moderate-PA and High-PA groups, the improvements in the forehead and glabellar lines compared to baseline were observed only after one month (*p* < 0.001, both) (Figure 1). As for the between-group comparisons, at the baseline assessment, the Low-PA group presented a significantly higher prevalence of “moderate lines” (45%) and a significantly lower prevalence of “very severe lines” (0%) compared with the other groups (*p* < 0.001). At the 1-month follow-up, no differences were found for “severe lines” and “very severe lines” for both areas. However, considering the 2nd and 3rd-month follow-up, the Low-PA group had a significantly lower prevalence of “severe lines” and “very severe lines” (*p* < 0.001) and glabellar lines (*p* < 0.002) compared to the Moderate-PA and High-PA groups. Further, no significant differences were found between Moderate-PA and High-PA groups in all subscales for forehead and glabellar lines throughout this study (*p* > 0.05) (Figure 1).

### 2.3. Patient Satisfaction with BoNT-A Treatment

For all groups, the intra-group assessment showed that satisfaction with BoNT-A treatment for lines between the eyebrows increased one month after treatment (*p* < 0.001). However, only the Low-PA group maintained high satisfaction throughout this study. The Moderate-PA group showed high satisfaction at the 1- and 2-month comparison (*p* < 0.001), while the High-PA group demonstrated increased values only at the 1-month follow-up (*p* < 0.001). Regarding between-group comparisons, higher values for patient satisfaction with treatment were found for the Low-PA group in the 2- and 3-month assessment, compared with the other groups (*p* < 0.001). Likewise, the Moderate-PA group had higher values compared with the High-PA group at the 2- and 3-month follow-ups (*p* < 0.01) (Table 2).

Considering the perceived age after BoNT-A treatment, the intra-group comparisons showed that in the Low and Moderate-PA groups, it decreased until the second assessment (*p* < 0.01). However, no statistically significant differences were found in the High-PA group in all periods of assessment (*p* > 0.05). Conversely, the between-group comparisons showed no significant differences in all follow-ups (*p* > 0.05) (Table 3).

## 3. Discussion

To the authors’ knowledge, this is the first clinical trial to assess the effects of physical activity on BoNT-A’s durability and to demonstrate the negative influence of high levels of physical activity on the duration of the esthetic effects. The results showed a significant reduction in muscle electrical activity in all groups, but this effect was only sustained throughout this study in the Low-PA group. Likewise, the severity of forehead and glabellar dynamic lines was significantly decreased at all timepoints only in the Low-PA group. Additionally, patient satisfaction with treatment for lines between eyebrows was significantly higher only in the Low- and Moderate-PA groups in all post-treatment follow-ups.

BoNT-A is a well-established treatment for facial esthetics, and recurrent users have reported that long-lasting results are absolutely essential when choosing a neuromodulator [13]. Therefore, assessing possible factors that can influence BoNT-A esthetic effects’ durability is of main importance. In this direction, our study assessed if the levels of physical activity could influence BoNT-A esthetic effect since patients–practitioners of physical exercises and clinicians perceived this relationship but without a scientific basis. Therefore, our study successfully demonstrated with subjective and objective assessments that the level of physical activity can certainly influence the duration of the BoNT-A effect when injected for esthetic purposes in the forehead and glabellar muscles since the groups with moderate and high physical activity presented a shorter effect of the neuromodulator in all variables assessed.

A significant reduction in electrical activity was found in all muscles after BoNT-A injections up to the first month of follow-up for all groups, with the Low-PA group having a greater reduction at the 1-month evaluation compared with the other two groups. Our results are in line with the current literature that shows that the maximum reduction of electrical activity after BoNT-A injections occurs in the first month after treatment [25,26,27].

Regarding subjective results, the Moderate-PA and High-PA groups had a higher prevalence of “severe” and “very severe” lines in the forehead and glabellar area from the second month onward compared with the Low-PA group. On the other hand, the Low-PA group had a lower prevalence of “severe and very severe lines” for the two areas at the 2- and 3-month follow-ups. Also, patients in the Low-PA and Moderate-PA groups were more satisfied with the treatment of lines between the eyebrows, with patients in the Low-PA groups being the most satisfied; however, this result was not reflected in the rating of perceived age, as patients in all groups perceived themselves younger after BoNT-A treatment throughout this study. In addition, for the High-PA group, younger perceived age was noted before treatment, and this did not change after treatment. Our subjective results agree with other clinical studies using different scales to assess BoNT-A improvement in the forehead and glabellar dynamic lines [8,28,29]. It is well known that the muscle paralysis effect of BoNT-A lasts up to 3 months; after a few days of muscle paralysis, a terminal sprouting process begins that, after a few weeks, begins to gradually restore muscle contraction [14]. In our study, the groups that had higher levels of physical activity showed muscle recovery already in the second month of assessment. In addition, Moderate-PA and High-PA groups had a lower electrical activity reduction and a higher prevalence of severe lines at the first month of assessment and at all timepoints compared with the Low-PA groups for all muscles.

Although there is no literature assessing the relationship between high levels of physical activity and the reduced duration of the BoNT-A effect, some mechanisms could explain our results. A recent literature review about treatments for dystonia reported that different modalities of rehabilitation therapy could normalize muscle activity, which include intensive motor training, passive and active external methods (biofeedback), and an association of neuromodulation and training, which aim to modify brain excitability, demonstrating that physical training could reestablish normal muscle contraction. However, they conclude that the evidence for this type of intervention was “very low” [30]. Also, the role of specific myokines relevant to the anabolic factor in muscle, such as insulin-like growth factor (IGF-1), should be considered in the reinnervation process of muscle tissue. Studies have shown that IGF-1 injections can reinnervate paralyzed muscles in experimental models of facial paralysis [23,24]. Despite conflicting results, IGF-1 levels have been shown to increase after physical exercise [22,24]. In addition, one study has shown that the local delivery of IGF-binding protein 4 (IGF-BP4) in muscles paralyzed by BoNT-A prevented nerve sprouting, concluding that muscle IGFs have a key role in terminal nerve sprouting [31]. Consequently, it could be hypothesized that the possible increase in IGF-1 levels due to intense exercise may be directly related to the lower duration of BoNT-A injections.

Despite the important findings of this study, some limitations should be mentioned. This study was performed without sex comparison, so our results cannot be extrapolated to the male population. Moreover, the doses applied to participants were not tailored to the severity of symptoms, which certainly would have affected the outcomes in patients that needed higher doses to achieve optimal esthetic results; however, since the main objective of our study was to assess the duration of BoNT-A effect, the doses had to be standardized. On the other hand, in the first month of assessment, all the participants were satisfied with the esthetic results, meaning that the doses were probably correct for their expectations. Also, it is important to mention that even though the severity of forehead and glabellar dynamic lines was significantly decreased at all timepoints, the Low-PA group presented with less “very severe lines” before treatment compared to the other groups, which could also influence the results. We recommend that future studies should assess the effects of other types of exercise since part of our population (Moderate- and High-PA) consisted only of CrossFit participants, which is considered high-intensity training. Likewise, future studies should use more accurate scales and objective evaluations to assess physical activity since the IPAQ is a subjective tool that cannot report data in a highly accurate manner; however, the literature has reported that the IPAQ instruments have reasonable measurement properties for monitoring population levels of physical activity among adults in diverse settings [32]. Additionally, some specific markers, like myokines related to nerve growth, should be assessed in this kind of patient.

As a final remark, our results are extremely important for clinical practice because they add knowledge about a specific factor that can reduce the durability of BoNT-A and permit us to recommend including the level of physical activity as an important topic to be addressed by clinicians in the anamneses of patients receiving esthetic injections of BoNT-A or for neuromuscular disorders. However, an important clinical question remains to be answered: do patients–practitioners of routine physical exercise need more BoNT-A in order to have a long-lasting esthetic effect?

## 4. Conclusions

Based on the results and limitations of this study, it can be concluded that higher levels of physical activity could have a negative influence on the durability of the esthetic effects of BoNT-A, diminishing its paralytic effect on frontalis, corrugator, and supercilii muscles and reducing the mean duration of the neuromodulator.

## 5. Materials and Methods

This was a single-center, controlled, single-blinded study. The study protocol was approved by the Research Ethics Committee of Uningá University, Paraná, Brazil (CAAE #59261322.0.0000.5220) and the Brazilian Registry of Clinical Trials (ReBEC—RBR-6ms6sns). All subjects were informed about the research purposes and provided a signed voluntary informed consent to participate in this study. This clinical trial was conducted in accordance with the ethical principles of the Declaration of Helsinki and the recommendations of the Consolidated Standards of Reporting Trials (CONSORT) guidelines. This study started after the approval of the Research Ethics Committee of Uningá University and the acceptance of the participants to be enrolled in the research.

### 5.1. Participants

This study included Brazilian women aged 25 to 50 years, practitioners or not (control group) of CrossFit, who complained of dynamic wrinkles and had moderate or severe dynamic forehead and glabellar lines according to the Merz 5-point scale [33]. Subjects were also required to have never received BoNT-A injections in any area of the face for esthetic or therapeutic indications, nor any other esthetic procedure for wrinkles. Exclusion criteria were antitetanic vaccination or any chemical peel in the past 6 months, procedures that could affect the forehead and glabellar regions in the previous 12 months, autoimmune disease, and/or current use of drugs that act at neuromuscular junctions.

### 5.2. Allocation and Blinding

Subjects included in this study (*n* = 60) were divided into three groups (*n* = 20) according to the Portuguese-Brazilian version of the short form of the International Physical Activity Questionnaire (IPAQ) [32,34], which assesses physical activity level in a comprehensive set of domains. Briefly, the short-form IPAQ captures four types of activity in four domains: walking, sitting, moderate-intensity activities, and vigorous-intensity activities. The total score [32,34] is calculated by summing the duration (in minutes) and frequency (days), classifying individuals into three levels of physical activity (PA): Low, Moderate, and High [35]. Participants allocated to the Low-PA group were considered the control group (no physical exercise of any kind, just daily activities), and participants of the Moderate and High-PA were practitioners of CrossFit training for 3 or 6 days per week (3 or 6 h), respectively. Subject allocation was performed by an independent researcher who was not involved in the assessment of the study outcomes. It is important to cite that even the participants in the Low-PA group had generally good health and a body mass index (BMI) considered normal (from 18.5 to 24.9).

### 5.3. Treatment

BoNT-A (100 U, Botox^®^, Allergan, Irvine, CA, USA) was reconstituted with 1.0 mL of preservative-free saline solution so that each 0.1 mL corresponded to 10 U of product. The doses applied in the frontalis, procerus, and corrugator supercili muscles followed the Italian consensus for onabotulinumtoxinA (Botox^®^) injections [10]. Therefore, 12 U, divided into 6 injection points, were applied in the frontalis muscle, a single 4-U injection was applied in the procerus muscle, and single 4U and 3U injections were applied, respectively, in the medial head and lateral part of the corrugator supercili muscle. The injected doses were not adjusted for individual esthetic needs. The injections were performed by the same researcher, who did not know the corresponding group of the patient and was not involved in any other process of this study. Injections were performed in a single session, and no compensatory injections were made in the case of not reaching an optimal result.

### 5.4. Outcomes

Subjects were evaluated at four timepoints (baseline, 30, 60, and 90 days) throughout the three months of study. Outcomes were assessed at all evaluation periods by a researcher not involved in any other procedure of this study.

#### 5.4.1. Primary Outcome

##### Electromyography Activity (EMG)

To record the EMG signal, an 8-channel EMG system (Miotool^®^, Sao Paulo, Brasil, 3 dB level EMG frequency range: 10–700 Hz; sampling rate 3000/s; resolution: 2.44 μV/bit;) was used by a single-calibrated operator (Kappa = 0.80). Bipolar surface electrodes, size 3.2 × 2.8 cm (Ag-AgCl disks, Covidien llc, Quebec, Canada), were fixed in the assessed muscles after the skin was cleaned with 70% alcohol, and the reference electrode was placed on the manubrium of the sternum. In the frontalis muscle, the electrodes were attached on the pupillary line, 2 cm above the eyebrow, in the procerus muscle at the cranial end of the nasal bones, and the corrugator supercili muscles at the midpoint of the cranial border of the eyebrow [36]. Then, participants were asked to perform the following facial expressions in front of a mirror to train their maximum voluntary contraction (MVC) for each facial expression:Forehead frowning (surprised facial expression).Glabellar frowning (angry upper facial expression).

To record the EMG signal in MVC, the subjects remained seated in a chair, with their head and shoulders held straight in a relaxed position and the Frankfurt plane parallel to the floor. Participants were asked to contract the assessed muscles as much as possible for 5 s to assess MVC. This procedure was repeated three times with a 1-minute rest between measurements to avoid muscle fatigue. The complete EMG signal was acquired at a frequency of 1000 Hz by using an analog-to-digital conversion device. Then, the EMG signal was subsequently band-pass filtered for 20–500 Hz to obtain the root-mean-square (RMS represents the square root of the average power of the EMG signal for a given period of time) value referring to 5 s of MVC of the facial muscles. The MiotecSuite software 1.0 (Miotec Equipamentos Biomédicos, Porto Alegre, Brazil) was used to analyze data acquired during MVC. The mean of the three recordings of MVC was used for statistical analyses.

#### 5.4.2. Secondary Outcomes

##### Severity of Forehead and Glabellar Lines

The severity of forehead and glabellar lines was assessed using the Merz 5-point scale [30] to rate the visible lines of the maximal frontalis, procerus, and corrugator supercilii muscles in the tense state as follows: 0 = no lines, 1 = mild lines, 2 = moderate lines, 3 = severe lines, 4 = very severe lines. The rating was conducted by a researcher who was not involved in other parts of this study (O.N.M.) and was based on a visual inspection of the patients.

##### Patient’s Perceived Satisfaction with BoNT-A Treatment

The FACE-Q questionnaire records patient-reported outcomes (PRO). This tool is composed of a series of scientifically validated independent assessment scales which measure the results of a range of facial procedures; however, it can be applied before facial procedures as well. The FACE-Q scales are validated scales for measuring outcome expectations and patient satisfaction at baseline and after treatment. Two Face-Q scales [37] were used to assess patient satisfaction with treatment: the Face-Q appraisal of lines between eyebrows [38], which assesses how much the participant is bothered by the area between her eyebrows over the past week on a 4-point scale—1 = not at all, 2 = a little, 3 = moderately, and 4 = extremely; and the FACE-Q patient-perceived age Visual Analogue Scale [39,40], with which participants report the age they perceive relative to their actual age prior to treatment. This scale ranges from 15 years older to 15 years younger, with increments of 1 year. To score the FACE-Q Appraisal of Lines Between Eyebrows scale, the items are summed to obtain a total raw score. Then, the conversion table (specific for each scale) is used to convert the raw score into a score that ranges from 0 (worst) to 100 (best). The authors signed a license agreement to use the FACE-Q scales for nonprofit academic research.

### 5.5. Statistical Analysis

Data for each group and timepoints are reported as means ± standard deviation (SD) or median, minimum, and maximum. The Shapiro–Wilk test was used to verify data distribution, which did not follow a normal distribution. EMG data were converted to logarithmic values, but the other variables did not show a normal distribution, even after conversion. Therefore, the two-way repeated measures ANOVA was used to compare intra- and intergroup differences over time for EMG, followed by the Sidak post hoc test. Non-parametric tests, such as the Friedman and Kruskal–Wallis, were used to compare differences over time and between groups, respectively, for all subjective scales. All analyses were performed using SPSS for Windows (21.0, SPSS Inc., Armonk, NY, USA), with a significance level of 5% using Bonferroni correction for multiple comparisons.

## Figures and Tables

**Figure 1 toxins-15-00463-f001:**
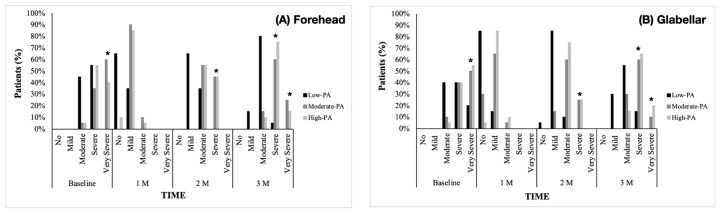
Changes in the Merz Esthetics Scale score for forehead (**A**) and glabellar dynamic lines (**B**) over time (%) after neuromodulation. PA: physical activity; M: month. * *p* < 0.05 between Low-PA group and the other groups.

**Table 1 toxins-15-00463-t001:** Changes (mean and standard deviation) in root mean square scores (RMS μV) after neuromodulation of the frontalis, corrugator supercilii, and procerus muscles at maximum voluntary contraction (MVC) over time.

Time				
*Frontalis*	Baseline	1 Month	2 Months	3 Months
Low PA	178.4 (±29.3)	46.9 (±19.8) *	81.9 (±21.4) *	130.9 (±23.9) *
Moderate PA	205.8 (±41.8)	109.5 (±52.5)	159.7 (±50.4)	191.5 (±45.4)
High PA	190.3 (±48.1)	85.4 (±48.1)	173.4 (±46.4)	188.3 (±47.0)
*Corrugator*				
Low PA	40.5 (±22.1)	10.1 (±4.8) *	20.0 (±10.5) *	29.6 (±15.0) *
Moderate PA	43.8 (±12.3)	18.6 (±8.4)	30.9 (±12.1)	39.0 (±12.4)
High PA	48.7 (±17.1)	20.0 (±12.7)	39.5 (±16.9)	45.3 (±17.0)
*Procerus*				
Low PA	32.9 (±18.8)	8.8 (±4.9) *	17.3 (±9.6) **	24.4 (±13.1) **
Moderate PA	31.5 (±7.14)	12.5 (±4.8)	22.5 (±8.5)	27.6 (±7.9)
High PA	35.3 (±11.2)	13.6 (±6.9)	29.0 (±11.0)	32.2 (±11.2)

* *p* < 0.05 between Low-PA group and the other groups. ** *p* < 0.05 between Low-PA group and High-PA group.

**Table 2 toxins-15-00463-t002:** Post-neuromodulation changes (Median, Min, and Max) in FACE-Q Appearance of Lines Between Eyebrows scores over time.

FQ-Eyebrows	Time			
	Baseline	1 Month	2 Months	3 Months
Groups	Median (Mn, Mx)	Median (Mn, Mx)	Median (Mn, Mx)	Median (Mn, Mx)
Low-PA	33 (21, 41) Aa	93 (87, 100) Ab	87 (81, 93) Ac	74 (63, 87) Ac
Moderate-PA	31 (15, 48) Aa	93 (81, 93) ABb	72 (59, 87) Bbc	48 (29, 55) Bc
High-PA	33 (21, 41) Aa	87 (59, 93) Bb	50 (25, 63) Cc	37 (21, 52) Bac

Different lowercase letters (horizontal) represent significant intra-group differences (*p* < 0.05). Different uppercase letters (vertical) represent significant between-group differences (*p* < 0.05).

**Table 3 toxins-15-00463-t003:** Post-neuromodulation changes (Median, Min, and Max) in FACE-Q Perceived Age Visual Analogue Scale over time.

FQ-AGE/VAS	Time			
	Baseline	1 Month	2 Months	3 Months
Groups	Median (Mn, Mx)	Median (Mn, Mx)	Median (Mn, Mx)	Median (Mn, Mx)
Low PA	0 (−7, 7) Aa	−5 (−10, −2) Ab	−4 (−8, −2) Ab	−3 (−7, 0) Aa
Moderate PA	0 (−8, 10) Aa	−4 (−15, 0) Ab	−3.5 (−15, 0) Ab	−2 (−12, 0) Aab
High PA	−3 (−10, 5) Aa	−4 (−15, 8) Aa	−4 (−15, 8) Aa	−3.5 (−15, 8) Aa

Different lowercase letters (horizontal) represent significant intra-group differences (*p* < 0.05). Different uppercase letters (vertical) represent significant between-group differences (*p* < 0.05).

## Data Availability

Datasets related to this article will be available upon request to the corresponding author.

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
