# Peer review of "High Levels of Physical Activity Reduce the Esthetic Durability of Botulinum Toxin Type A: A Controlled Single-Blind Clinical Trial"

_toxins, 2023, doi:10.3390/toxins15070463_

Round 1

Reviewer 1 Report

In this study, the authors found that increased physical activity leads to a reduced duration of action after BoNT treatment in the face. This is an interesting observation, however, the explanation is weak and not really convincing.

My comments:

Please provide some more details on the intensity of physical activity in the 3 subgroups. Which kind of activity, duration, body part trained? How was the training status of participants before injection (intensity and duration)? Maybe there is a difference between chronically trained people and those with just recently started activity.

As BoNT is frequently injected in movement disorders, such as dystonias or spasticity, this observation should be true for all indications of BoNT. Has this ever been observed in other indications? What is the rationale behind the authors´ hypothesis? Is it just based on subjective impressions reported by patients?

The effect of physical activity on muscles previously injected has been examined with heterogeneous results. How do the authors explain that exercise in (healthy) muscles (far) distant from injected muscles may have an influence on duration of remote muscles?

The results of ref. 18 and 19 refer to the situation after spinal cord lesions. This is completely different from this work where healthy people without any neurological disorder were injected. Moreover, the authors should not mix up denervation/reinnervation after nerve lesions with the restorage of the endplate after BoNT treatment. In the latter, there is transient sprouting which is not the same as reinnervation after nerve lesions. Therefore, it is difficult to argue with the results of theses studies. The same is true for IGF-1 application.

see above

Author Response

Reviewer 1

In this study, the authors found that increased physical activity leads to a reduced duration of action after BoNT treatment in the face. This is an interesting observation, however, the explanation is weak and not really convincing.

Thank you for handling our manuscript. We revised it according to your comments and addressed each issue bellow, in a point-by-point manner. Our modifications are in track changes in the manuscript. We feel that by incorporating your suggestions in the revised version the clarity of our manuscript has improved.

Comments:

- Please provide some more details on the intensity of physical activity in the 3 subgroups. Which kind of activity, duration, body part trained? How was the training status of participants before injection (intensity and duration)? Maybe there is a difference between chronically trained people and those with just recently started activity.                           Reply: Thank you for this observation. The participants of the moderate and high physical activity (PA) groups were practitioners of Crossfit training, which uses high intensity functional movements. The participants of the moderate PA group reported training 3 days per week, and the high PA group 6 days per week. The total number of hours was of 3 or 6 hours per week, according to the group. The participants of the control group (Low-PA) did not practice crossfit or any kind of physical exercise. It is important to mention that the IPAQ assesses all the physical activities in participant’s everyday life, such as cleaning the house, walking, sitting.

Reviewed text: Page 6, lines 213-217 and Page 7, lines 262-264.

- As BoNT is frequently injected in movement disorders, such as dystonias or spasticity, this observation should be true for all indications of BoNT. Has this ever been observed in other indications? What is the rationale behind the authors´ hypothesis? Is it just based on subjective impressions reported by patients?                                                   Reply: Thanks for the question. As far as we know, this is the first clinical study assessing the effects of physical activity on BoNT-A’s durability, irrespective of the purpose of its application. The rationale of our hypothesis is based on literature reporting that the muscular contractions of the mimic muscles during the period of intense physical activity due to the force exerted during this period, could accelerate reinnervation of the paralyzed muscle. Additionally, even though there is no consensus about the increase of biomarkers related with reinnervation after intense physical activity, this could be the main hypothesis. Finally, the reduced duration of BoNT-A in patients with intense routine of physical activity is a recurrent observation of clinicians and patients’ complaint. Therefore, it deserves to be investigated.

- The effect of physical activity on muscles previously injected has been examined with heterogeneous results. How do the authors explain that exercise in (healthy) muscles (far) distant from injected muscles may have an influence on duration of remote muscles? Reply: Thanks for the suggestion. Unfortunately, we do not have this answer and did not assess this variable. But it is important to mention, that patients that practice intense physical exercises also contract the muscles of the face, especially the mimic muscles, which was the target of our study. Therefore, this contraction could be considered a stimulus to quickly reinnervate the muscles.

- The results of ref. 18 and 19 refer to the situation after spinal cord lesions. This is completely different from this work where healthy people without any neurological disorder were injected. Moreover, the authors should not mix up denervation/reinnervation after nerve lesions with the restorage of the endplate after BoNT treatment. In the latter, there is transient sprouting which is not the same as reinnervation after nerve lesions. Therefore, it is difficult to argue with the results of these studies. The same is true for IGF-1 application.                                                     Reply: We appreciate your relevant comment and agree that the references 18 and 19 as well as those related to IGF-1 are regarding a different subject (denervation/reinnervation). However, the mechanism by which intense physical activity promotes increased nerve sprouting has never been explored (considering BoNT-A durability). Therefore, we have no other literature available to explain the possible influence of exercise, except for these papers affirming that practicing intense physical activity generate new motor nerve sprouts. As researchers, it is our role to make hypothesis and suggest future studies to confirm those. Nevertheless, to better clarify that the cited references are hypothesis (and not a definitive mechanism of action), we reformulated the sentences.

Reviewer 2 Report

This study investigated the effects of physical activity on duration of aesthetic action of BoNT/A in female patients and concludes that patients that do not exercise regularly respond better to BoNT/A injections for improvement of forehead or glabellar wrinkles. The findings are novel, interesting, and potentially could be helpful to clinicians in aesthetic practice. However, some potential limitations in the study are not identified. Also, the Discussion and interpretation could be more balanced.

Specific points

The title should reflect the outcome of the study, rather than question investigated. It should also make it clear that it is type A botulinum toxin that was used.

L6-8 The Abstract should make it known that it was onabotulinumtoxinA that was used in this study.

L42 Note that the timescale quoted is specifically referring to the aesthetic effects of BoNT/A

L43-44 The statement is only partly correct. Full functional recovery is associated with restoration of neurotransmission at the original motor nerve endplates. During the transition from partial to full recovery, as neurotransmission progressively returns at the original endplate, the terminal sprouts retract and eventually disappear (unless the muscle is paralysed again). Also, L166-168, L180 and L183 the statements are inaccurate. BoNT/A does not cause muscle denervation but induces paralysis by blockade of neurotransmission. The innervating nerves and motor endplates remain morphologically intact throughout the period of muscle paralysis and functional recovery. During functional recovery, muscles are not re-innervated, but the endplates elaborate neurites in a process known as terminal sprouting. In mouse sternomastoid muscle, such sprouting is not delayed until after 3 months of muscle paralysis, it initiates within days and within weeks the sprouts form junctions with the muscles where neurotransmission can apparently take place and support a partial recovery of muscle activity.

L48-50 Indicate whether these factors are specific to the aesthetic use of BoNT/A, or apply more generally.

L55 & L177 The statements are misleading as the cited references describe sprouting by nerves in the spinal cord, some of which occurs around motor nerve cell bodies. They did not report new motor nerve sprouts.

L70-72 Given that the study participants were assigned to different groups based on their levels of physical activity, it seems probable that the groups differed in terms of general fitness and health, and that this could be a factor in the outcomes. In this regard, it would be informative to document any differences in average weight and body mass index between the groups.

L75 It is not clear how the probability was calculated or what exactly is being compared.

Figure 1 Error bars should be plotted. It is stated that data and time points for each group are reported as mean ± standard deviation (L305) but the SD is not plotted in any of the figures. The figures were reproduced very small in the PDF file provided and could only be read at very high magnification on screen, which is not ideal as the resolution of the images was compromised.

L92 It is not explained precisely what data the statistical test was applied to.

L90-102 & Figure 2 It is a major weakness of the study that the baseline esthetic scores for the low physical activity group are less severe than those for the moderate- and high-PA groups. This needs to be reported more explicitly and the implications for interpretation of the scores after BoNT/A need to be discussed (L131-140; L158-160) in a more balanced way. As the scores for the low-PA group are lower before treatment, this must be considered as a factor in the lower scores for the same group at all stages after of follow up after treatment. Simply put, milder symptoms are easier to treat. In this context, discussion is warranted in regard to the relationship between milder aesthetic scores at baseline and lower EMG activity at baseline for the low PA group compared to moderate- and high-PA in the frontalis (Figure 1A) and corrugator muscles (Figure 1B), and compared to high-PA in the procerus (Figure 1C). Can the authors explain the lower muscle activity and aesthetic scores at baseline for the low-PA group compared to the others? Are they simply more relaxed?

L145 The sentence is difficult to follow ‘both patients practitioners of physical exercises and clinicians’. Does this mean that the patients are the practitioners of physical exercises, or is the sentence listing three groups i.e. patients, practitioners of physical exercise and clinicians?

L145-146 Cite the source of this information. Is it the personal experience(s) of the author(s) or anecdotal evidence from some other source? How could patients know that exercise reduces the effect? Is there evidence that patients who exercise more return for secondary treatment sooner than those that don’t?

L159-160 This statement is not supported by Figure 2. The low -PA group had a higher prevalence of ‘no lines’ for both groups at 1 month follow-up only, and for glabellar at 2 months but not 3 months.

L168-169 It is not clear what ‘this pattern’ refers to. The preceding sentence describes muscle denervation and re-innervation after BoNT/A (an inaccurate description, as pointed out in previous comments) but this study only examined muscle electrical activity.

L178-189 The authors make an interesting argument regarding the possible role of exercise in inducing IGF-1 that can enhance or accelerate motor nerve sprouting and, possibly thereby, functional muscle recovery. However, they need to consider the evidence that BoNT/A-induced muscle paralysis itself induces IGF expression (Ishii, D. 1989, PNAS 86, 2898-2902) and that administration of IGF sequestering proteins to BoNT/A paralysed muscles suppresses terminal sprouting (Caroni, P. et al., 1994, J. Cell Biol. 125, 893-902). It is also pertinent to cite the evidence that paralysis of the gastrocnemius muscle in rat with BoNT/A does not block exercise-induced increases in the growth of this muscle.

L209-213 An alternative conclusion is that the low-PA cohort in this study tended to have less severe muscle hyperactivity and glabellar/forehead lines than the moderate- and high-PA groups before treatment and this is why they responded more positively to BoNT/A.

L256, L281 Were the assessors blinded to the treatment groups?

Spelling errors

L21, L145, L208 esthetic (use American spelling throughout)

L22 diminished

L35 frequently

L143 esthetics

L146 clinicians

L203 a specific

L207 routinely

L207 long-lasting

L228 therapeutic

The quality of English language is very good and the paper is well written, but a careful spell check is recommended. I have highlighted some errors spotted, but I probably have not detected all of them.

Author Response

Reviewer 2:

This study investigated the effects of physical activity on duration of aesthetic action of BoNT/A in female patients and concludes that patients that do not exercise regularly respond better to BoNT/A injections for improvement of forehead or glabellar wrinkles. The findings are novel, interesting, and potentially could be helpful to clinicians in aesthetic practice. However, some potential limitations in the study are not identified. Also, the Discussion and interpretation could be more balanced.

 - The title should reflect the outcome of the study, rather than question investigated. It should also make it clear that it is type A botulinum toxin that was used.               Reply: Thanks for the suggestion. The title was reformulated.                                  Revised text: Page 1, lines 2-3.

- L6-8 The Abstract should make it known that it was onabotulinumtoxinA that was used in this study.                                                                                                                        Reply: Thanks for the suggestion. This information was added to the Abstract. Revised text: Page 1, line 19.

- L42 Note that the timescale quoted is specifically referring to the aesthetic effects of BoNT/A.                                                                                                                      Reply: Thanks for this observation. This information was added to the manuscript. Revised text: Page 2, line 56.

- L43-44 The statement is only partly correct. Full functional recovery is associated with restoration of neurotransmission at the original motor nerve endplates. During the transition from partial to full recovery, as neurotransmission progressively returns at the original endplate, the terminal sprouts retract and eventually disappear (unless the muscle is paralysed again). Also, L166-168, L180 and L183 the statements are inaccurate. BoNT/A does not cause muscle denervation but induces paralysis by blockade of neurotransmission. The innervating nerves and motor endplates remain morphologically intact throughout the period of muscle paralysis and functional recovery. During functional recovery, muscles are not re-innervated, but the endplates elaborate neurites in a process known as terminal sprouting. In mouse sternomastoid muscle, such sprouting is not delayed until after 3 months of muscle paralysis, it initiates within days and within weeks the sprouts form junctions with the muscles where neurotransmission can apparently take place and support a partial recovery of muscle activity.                     Reply: Thank you for your comment. We agree with you and was not our intention to suggest that BoNT-A injection causes muscle denervation. We have used such references once it is the one that better explain nerve/muscle alterations related to physical exercise. We have adjusted the manuscript and hope that it is clearer now.

L48-50 Indicate whether these factors are specific to the aesthetic use of BoNT/A or apply more generally.                                                                                                     Reply: Thank you for the suggestion. This information was added to the manuscript. Revised text: Page 2, line 63.

- L55 & L177 The statements are misleading as the cited references describe sprouting by nerves in the spinal cord, some of which occurs around motor nerve cell bodies. They did not report new motor nerve sprouts.                                                                   Reply: We appreciate your observation. We have adjusted the manuscript accordingly. Revised text: Page 2, line 56.

L70-72 Given that the study participants were assigned to different groups based on their levels of physical activity, it seems probable that the groups differed in terms of general fitness and health, and that this could be a factor in the outcomes. In this regard, it would be informative to document any differences in average weight and body mass index between the groups.                                                                                                       Reply: It is an interesting observation. It is important to emphasize that the participants in the low physical activity group were health and had the body mass index (BMI) considered normal. This information was included in the Material and Methods section.

L75 It is not clear how the probability was calculated or what exactly is being compared.       Reply: We apologize, but we could not understand which probability is being cited by the reviewer. Could you clarify this question?

Figure 1 Error bars should be plotted. It is stated that data and time points for each group are reported as mean ± standard deviation (L305) but the SD is not plotted in any of the figures. The figures were reproduced very small in the PDF file provided and could only be read at very high magnification on screen, which is not ideal as the resolution of the images was compromised.                                                                                       Reply: We apologize that it was difficult to read our images. We tried to make Figures with SD, but it did not provide a good visual to readers. Therefore, we decided to replace the Figures by one Table.

L92 It is not explained precisely what data the statistical test was applied to.                    Reply: We apologize that it was not clear. The two-way repeated measures ANOVA was used to compared intra- and intergroup differences over time for EMG, followed by Sidak post-hoc test. Non-parametric tests, such as Friedman and Kruskal-Wallis, were used to compared differences over time and between groups, respectively, for all subjective scales Revised text: Page 9.

- L90-102 & Figure 2 It is a major weakness of the study that the baseline esthetic scores for the low physical activity group are less severe than those for the moderate- and high-PA groups. This needs to be reported more explicitly and the implications for interpretation of the scores after BoNT/A need to be discussed (L131-140; L158-160) in a more balanced way. As the scores for the low-PA group are lower before treatment, this must be considered as a factor in the lower scores for the same group at all stages after of follow up after treatment. Simply put, milder symptoms are easier to treat. In this context, discussion is warranted in regard to the relationship between milder aesthetic scores at baseline and lower EMG activity at baseline for the low PA group compared to moderate- and high-PA in the frontalis (Figure 1A) and corrugator muscles (Figure 1B) and compared to high-PA in the procerus (Figure 1C). Can the authors explain the lower muscle activity and aesthetic scores at baseline for the low-PA group compared to the others? Are they simply more relaxed?                                                                               Reply: It is an important consideration, and we appreciate that you had read our manuscript with such careful attention. Considering the Merz scale we found this difference in the wrinkles’ characteristics. However, as part of our inclusion/exclusion criteria, moderate wrinkles would already be notable and could cause complaints. Conversely, considering the EMG results, despite the mean of the Low-PA group was lower, it was not statistically significant different from the other groups. Therefore, we should not consider them as ‘more relaxed’, since the RMS μV are not different from the other groups. It has to be pointed out that the patients with more severe wrinkles should receive higher doses. However, since it is a research protocol, we had to standardize the doses irrespective of the group. This important point is cited in the Discussion section, as a limitation of the study. (page 6, highlighted in yellow)

- L145 The sentence is difficult to follow ‘both patients practitioners of physical exercises and clinicians’. Does this mean that the patients are the practitioners of physical exercises, or is the sentence listing three groups i.e. patients, practitioners of physical exercise and clinicians?                                                                                                                    Reply: Thanks for the suggestion. The sentence was corrected. We referred patients as practitioners of physical exercises too.                                                             Revised text: Page 5, line165.

- L145-146 Cite the source of this information. Is it the personal experience(s) of the author(s) or anecdotal evidence from some other source? How could patients know that exercise reduces the effect? Is there evidence that patients who exercise more return for secondary treatment sooner than those that don’t?                                                         Reply: Thanks for the questions. This information is based on reports of clinicians and patients; however, no scientific evidence has supported these reports.             Revised text: Page 2, line 56.

- L159-160 This statement is not supported by Figure 2. The low -PA group had a higher prevalence of ‘no lines’ for both groups at 1 month follow-up only, and for glabellar at 2 months but not 3 months.                                                                                                    Reply: Thanks for the suggestion. This information was corrected accordingly to the results of Figure 2.                                                                                                            Revised text: Page 5, line 180.

- L168-169 It is not clear what ‘this pattern’ refers to. The preceding sentence describes muscle denervation and re-innervation after BoNT/A (an inaccurate description, as pointed out in previous comments) but this study only examined muscle electrical activity.                                                                                                             Reply: Thanks for this observation. “This pattern” was related to the duration of botulinum toxin injection (3 months). We agree with the reviewer and have rewritten the sentence to avoid misinterpretation regarding BoNT-A effect on muscles.

- L178-189 The authors make an interesting argument regarding the possible role of exercise in inducing IGF-1 that can enhance or accelerate motor nerve sprouting and, possibly thereby, functional muscle recovery. However, they need to consider the evidence that BoNT/A-induced muscle paralysis itself induces IGF expression (Ishii, D. 1989, PNAS 86, 2898-2902) and that administration of IGF sequestering proteins to BoNT/A paralysed muscles suppresses terminal sprouting (Caroni, P. et al., 1994, J. Cell Biol. 125, 893-902). It is also pertinent to cite the evidence that paralysis of the gastrocnemius muscle in rat with BoNT/A does not block exercise-induced increases in the growth of this muscle.                                                                                        Reply: Thank you for suggesting such interesting references. It is exactly the point of our discussion. The cited references were included in the manuscript.

- L209-213 An alternative conclusion is that the low-PA cohort in this study tended to have less severe muscle hyperactivity and glabellar/forehead lines than the moderate- and high-PA groups before treatment and this is why they responded more positively to BoNT/A.                                                                                                                Reply: Thanks for the suggestion. Even though we partially agree with the reviewer about the fact that the low-PA group presented less severe muscle activity, our electromyographic data, which is an objective assessment for muscle paralysis, showed no significant differences between groups at baseline.        Therefore, the conclusion of the manuscript refers to the effect of physical exercise on the durability of BoNT-A, which is supported by the data of the Moderate and High-PA group.

L256, L281 Were the assessors blinded to the treatment groups?                            Reply: Yes, they were blinded. This information was mentioned in the Allocation and Blinding and treatment section.                                                                   Revised text: Page 7, lines 286-288; 297-299 and 303-304.

- Spelling errors

L21, L145, L208 esthetic (use American spelling throughout)

L22 diminished

L35 frequently

L143 esthetics

L146 clinicians

L203 a specific

L207 routinely

L207 long-lasting

L228 therapeutic

Reply: Thanks for the corrections. The English language was double checked throughout the manuscript.

Reviewer 3 Report

The article is well written and the novelty of the research is very intriguing.

However, I have some concerns:

-line53:please add reference

- line231: add neuromuscular disorders

- short- form of IPAQ: this is a qualitative scale in which the patient self-reports his/her physical activity. The score obtained could be different from reality; please comment the BIAS and implement the limitation of the study. More accurate scales and evaluations should be made to assess the physical activity.

- The authors used surface electrodes; I have some doubt about the reliability. Moreover, the signal is dependent on the muscle activity; how the authors checked the maximal contraction of these patients? How was the signal assessed in detail? please explain RMS (Root mean square). This is an important methodological factor; the authors should provide all the elements needed to reproduce this study in detail (scientific method) because it is not clear.

- please provide high quality figures; moreover, please make the figures larger, they are difficult to read.

Author Response

Reviewer 3

The article is well written, and the novelty of the research is very intriguing.

However, I have some concerns:

Reply: Thank you for your careful read and considerations about our manuscript. We hope that all modifications meet your requirements.

-line53:please add reference.

Reply: We would like to include a reference, but no previous paper has reported it. The motivation of this research is based in clinical observation.

- line231: add neuromuscular disorders

Reply: Thanks for the suggestion. The text was added to the manuscript.

- short- form of IPAQ: this is a qualitative scale in which the patient self-reports his/her physical activity. The score obtained could be different from reality; please comment the BIAS and implement the limitation of the study. More accurate scales and evaluations should be made to assess the physical activity.

Reply: Thanks for the suggestions, we agree with the reviewer. Nevertheless, it is important to mention that the IPAQ instruments have acceptable measurement properties, are at least as good as other established self-reports and has reasonable measurement properties for monitoring population levels of physical activity among 18- to 65-yr-old adults in diverse settings. "Craig CL, Marshall AL, Sjöström M, Bauman AE, Booth ML, Ainsworth BE, Pratt M, Ekelund U, Yngve A, Sallis JF, Oja P. International physical activity questionnaire: 12-country reliability and validity. Med Sci Sports Exerc. 2003 Aug;35(8):1381-95. doi: 10.1249/01.MSS.0000078924.61453.FB. PMID: 12900694.” Revised text: Page 6 lines 233-238

- The authors used surface electrodes; I have some doubt about the reliability. Moreover, the signal is dependent on the muscle activity; how the authors checked the maximal contraction of these patients? How was the signal assessed in detail? please explain RMS (Root mean square). This is an important methodological factor; the authors should provide all the elements needed to reproduce this study in detail (scientific method) because it is not clear.

Reply: We apologize for the absence of information. The text was modified to better described all relevant information. Revised text: Page 8, lines 331-341

- please provide high quality figures; moreover, please make the figures larger, they are difficult to read.

Reply: We apologize for the quality of the Figures. They have been submitted as separate files with 300dpi to ensure that they will be easier to read.

Round 2

Reviewer 1 Report

Thank you for the reply, however, I still have comments.

Again, physical activity after spinal cord lesions cannot be compared with botulinum toxin applications. I recommend to delete this part. In fact, there are several publications on repetitive nerve stimulation after BTX in spasticity and dystonia. Why did the authors not include these papers in their discussion which would add to the discussion, in particular as these results were heterogeneous and in part different form the authors´ observations. In view of that I would recommend to give the discussion a more cautious tenor and formulate the results as a hypothesis instead of a clear fact. 

.

Author Response

Reviewer 1

Thank you for the reply, however, I still have comments.

Reply: Thank you for handling our manuscript. Please, see below the answer to the new suggestions.

Again, physical activity after spinal cord lesions cannot be compared with botulinum toxin applications. I recommend to delete this part. In fact, there are several publications on repetitive nerve stimulation after BTX in spasticity and dystonia. Why did the authors not include these papers in their discussion which would add to the discussion, in particular as these results were heterogeneous and in part different form the authors’ observations. In view of that I would recommend to give the discussion a more cautious tenor and formulate the results as a hypothesis instead of a clear fact. 

Reply: Thanks for the pertinent suggestion. We have deleted from the text all information regarding the physical activity after spinal cord lesions, as well as had included in the 6 paragraph in the Discussion section considering new studies about nerve stimulation in dystonia.

Further, regarding formulating the Results as a hypothesis, our results were findings from the research. Even though we understand the reviewer’s concerns, the Results are something concrete, not hypothesis. However, we understand that most of the mechanisms are not well elucidated, so we raised some hypothesis in the Discussion and Conclusion sections.

Reviewer 2 Report

The authors are thanked for their detailed responses to my comments. The revised manuscript is much better, but not all my comments have been rebutted adequately. Below, page and line references refer to viewing the document without showing tracked changes.

The studies [18,19] on the effect of exercise after spinal cord lesion did not report motor nerve sprouting as they only examined axonal regeneration and collateral sprouting of spinal nerves. This spinal nerve regeneration is not equivalent to the recovery of neurotransmission at motor nerve-muscle terminals after blockade of neuromuscular transmission with BoNT/A. Statements in the text are misleading and should be deleted/modified (see below). They are not critical to understanding the results because they relate only to speculation by the authors on a possible mechanism for muscle function recovery, a hypothesis that this study does not directly address experimentally.

Page 2 ln56. Delete “Experimental studies have shown that physical activity before or after denervation processes promotes the formation of new motor nerve sprouts, allowing for the recovery of muscle contraction [18,19].”

Page 5 ln182 replace “generate new motor nerve sprouts that allow” with ‘enhances’ and change “improve” to ‘improves’.

Page 1 Ln44 change to  ‘The waning of the effect is mainly related to the sprouting of new nerve endings, followed by recovery of function at the original nerve terminal and retraction of the sprouts’ (see https://doi.org/10.1073/pnas.96.6.3200)

Page 5 Ln174 replace “after that, the terminal sprouting process takes place causing the restoration of muscle contraction” with ‘after a few days of muscle paralysis a terminal sprouting process begins that, after a few weeks, begins to gradually restore muscle contraction’

The authors provided a very good response to comments on the less severe symptoms in the low-PA group at baseline, valid points that should be included in the Discussion. For example, p5 ln 142 where it states “the severity of forehead and glabellar dynamic lines was significantly decreased at all time-points only in the Low-PA group” it should be noted that the low-PA group had less “very severe” lines before treatment.

Ln 146 it should be noted that for the high-PA group a younger perceived age was noted before treatment and this did not change after treatment.

Ln165 It should be noted that Low PA group also had much less “very severe” lines than the other groups before treatment.

I am still unclear how some of the probability values were determined.

Results section – EMG – Ln78 (p<0.0001) What exactly is being compared? There are multiple groups and multiple time-points. It seems like each group is being compared at various times after treatment to the pre-treatment baseline, but is each group being compared individually and all produce a result of p<0.0001? Are all follow-up times being included in the calculation together, or is each time-point compared to baseline individually and always producing a result of p<0.0001?

Results section – Severity of forehead and glabellar lines – Ln98 (p<0.001) In this case there are three groups (high-, medium- and low-PA), two measurements (forehead and glabellar lines) and multiple time points (baseline, 1, 2 and 3 months). What exactly is being compared?

Results section – Patient satisfaction with BoNT-A treatment – Ln115 (p<0.001) Here it is clear that data at 1 month post-treatment is compared to pre-treatment (baseline), but there are three groups (high-, medium- and low-PA). However, Table 2 indicates p<0.05 for each group for such comparisons.

Results section – Patient satisfaction with BoNT-A treatment – Line Ln128 (p<0. 01) Two groups, multiple time-points.

For each of the above examples, it would be clearer if the in addition to the p values it could be indicated inside the brackets the statistical test used and the data groups being compared, or if multiple comparisons were performed and always produced the same result.

Other comments

Page 2 Results Ln74 Briefly detail here the assignment of subjects to groups and define the abbreviations used for these groups. These are referred to throughout the Results but not described until the Materials and Methods, which is at the end of the report.

Page 3 Ln100 Fig 2 is now Fig 1.

Page 5 Ln 150 ‘its’ should be ‘is’ – ‘durability is of main importance’.

Page 5 Ln 152 delete comma ‘clinicians perceived’

Page 6 Ln196 after ‘tailored’ insert ‘to the severity of symptoms’

Page 6 Ln 215 change to ‘practitioners of routinary physical exercise’

Only minor editing required.

Author Response

Reviewer 2:

The authors are thanked for their detailed responses to my comments. The revised manuscript is much better, but not all my comments have been rebutted adequately. Below, page and line references refer to viewing the document without showing tracked changes.

Reply: We would like to thank the reviewer for the detailed correction and reading of our study. We fill that by adopting the reviewer’s corrections the manuscript improved a lot. Please find below the answers to the new suggestions.

The studies [18,19] on the effect of exercise after spinal cord lesion did not report motor nerve sprouting as they only examined axonal regeneration and collateral sprouting of spinal nerves. This spinal nerve regeneration is not equivalent to the recovery of neurotransmission at motor nerve-muscle terminals after blockade of neuromuscular transmission with BoNT/A. Statements in the text are misleading and should be deleted/modified (see below). They are not critical to understanding the results because they relate only to speculation by the authors on a possible mechanism for muscle function recovery, a hypothesis that this study does not directly address experimentally.

Reply: We would like to thank the reviewer for this important suggestion. We deleted from the Discussion section all the text related to these two studies. Further, new studies were included to better discuss our results.

Page 2 ln56. Delete “Experimental studies have shown that physical activity before or after denervation processes promotes the formation of new motor nerve sprouts, allowing for the recovery of muscle contraction [18,19].”

Reply: The text was deleted.

Page 5 ln182 replace “generate new motor nerve sprouts that allow” with ‘enhances’ and change “improve” to ‘improves’.

Reply: The text was replaced.

Page 1 Ln44 change to  ‘The waning of the effect is mainly related to the sprouting of new nerve endings, followed by recovery of function at the original nerve terminal and retraction of the sprouts’ (see https://doi.org/10.1073/pnas.96.6.3200)

Reply: The text was changed.

Page 5 Ln174 replace “after that, the terminal sprouting process takes place causing the restoration of muscle contraction” with ‘after a few days of muscle paralysis a terminal sprouting process begins that, after a few weeks, begins to gradually restore muscle contraction’

Reply: The text was replaced.

The authors provided a very good response to comments on the less severe symptoms in the low-PA group at baseline, valid points that should be included in the Discussion. For example, p5 ln 142 where it states “the severity of forehead and glabellar dynamic lines was significantly decreased at all time-points only in the Low-PA group” it should be noted that the low-PA group had less “very severe” lines before treatment.

Reply: This information was added in the paragraph in which the limitations of the study were reported (last paragraph – Discussion section).

Ln 146 it should be noted that for the high-PA group a younger perceived age was noted before treatment and this did not change after treatment.

Reply: This information was added in paragraph 4, Ln 146 - Discussion section.

Ln165 It should be noted that Low PA group also had much less “very severe” lines than the other groups before treatment.

Reply: This information was added in the paragraph in which the limitations of the study were reported (last paragraph – Discussion section).

I am still unclear how some of the probability values were determined.

Results section – EMG – Ln78 (p<0.0001) What exactly is being compared? There are multiple groups and multiple time-points. It seems like each group is being compared at various times after treatment to the pre-treatment baseline, but is each group being compared individually and all produce a result of p<0.0001? Are all follow-up times being included in the calculation together, or is each time-point compared to baseline individually and always producing a result of p<0.0001?

Reply: Thank you for your valuable observation. In this first sentence, it is demonstrated the intragroup comparison, i.e., baseline compared to 1 month, 2 months and 3 months. All cited muscles were tested in all timepoints using the mixed design ANOVA test with the SPSS software. Please, see below the results (the number 1 in the box below is the baseline data; number 2, 1 month; number 3, 2 months; number 4, 3 months):

Frontalis

Comparações de pares

Medida:   MEASURE_1 

(I) Tempo

(J) Tempo

Diferença média (I-J)

Modelo padrão

Sig.b

Intervalo de confiança 95% para a diferençab

Limite inferior

Limite superior

1

2

,438*

,023

,000

,374

,502

3

,171*

,011

,000

,140

,201

4

,055*

,006

,000

,038

,072

2

1

-,438*

,023

,000

-,502

-,374

3

-,267*

,020

,000

-,322

-,212

4

-,383*

,022

,000

-,443

-,323

3

1

-,171*

,011

,000

-,201

-,140

2

,267*

,020

,000

,212

,322

4

-,116*

,008

,000

-,137

-,095

4

1

-,055*

,006

,000

-,072

-,038

2

,383*

,022

,000

,323

,443

3

,116*

,008

,000

,095

,137

Baseado em médias marginais estimadas

*. A diferença média é significativa no nível ,05.

b. Ajustamento para comparações múltiplas: Bonferroni.

Procerus

Comparações de pares

Medida:   MEASURE_1 

(I) Tempo

(J) Tempo

Diferença média (I-J)

Modelo padrão

Sig.b

Intervalo de confiança 95% para a diferençab

Limite inferior

Limite superior

1

2

,481*

,024

,000

,416

,546

3

,178*

,014

,000

,141

,216

4

,079*

,011

,000

,048

,109

2

1

-,481*

,024

,000

-,546

-,416

3

-,302*

,023

,000

-,364

-,241

4

-,402*

,023

,000

-,466

-,338

3

1

-,178*

,014

,000

-,216

-,141

2

,302*

,023

,000

,241

,364

4

-,100*

,009

,000

-,126

-,074

4

1

-,079*

,011

,000

-,109

-,048

2

,402*

,023

,000

,338

,466

3

,100*

,009

,000

,074

,126

Baseado em médias marginais estimadas

*. A diferença média é significativa no nível ,05.

b. Ajustamento para comparações múltiplas: Bonferroni.

Corrugator supercilii

Comparações de pares

Medida:   MEASURE_1 

(I) Tempo

(J) Tempo

Diferença média (I-J)

Modelo padrão

Sig.b

Intervalo de confiança 95% para a diferençab

Limite inferior

Limite superior

1

2

,483*

,024

,000

,419

,548

3

,185*

,013

,000

,150

,220

4

,069*

,008

,000

,046

,091

2

1

-,483*

,024

,000

-,548

-,419

3

-,299*

,023

,000

-,362

-,236

4

-,415*

,025

,000

-,482

-,347

3

1

-,185*

,013

,000

-,220

-,150

2

,299*

,023

,000

,236

,362

4

-,116*

,011

,000

-,146

-,086

4

1

-,069*

,008

,000

-,091

-,046

2

,415*

,025

,000

,347

,482

3

,116*

,011

,000

,086

,146

Baseado em médias marginais estimadas

*. A diferença média é significativa no nível ,05.

b. Ajustamento para comparações múltiplas: Bonferroni.

After, in the same paragraph, we demonstrated the between-group comparison data (Low-PA x Moderate-PA x High-PA). We had added the words intra-group and between-group to clarify this information to readers.

Results section – Severity of forehead and glabellar lines – Ln98 (p<0.001) In this case there are three groups (high-, medium- and low-PA), two measurements (forehead and glabellar lines) and multiple time points (baseline, 1, 2 and 3 months). What exactly is being compared?

Results section – Patient satisfaction with BoNT-A treatment – Ln115 (p<0.001) Here it is clear that data at 1 month post-treatment is compared to pre-treatment (baseline), but there are three groups (high-, medium- and low-PA). However, Table 2 indicates p<0.05 for each group for such comparisons.

Results section – Patient satisfaction with BoNT-A treatment – Line Ln128 (p<0.01) Two groups, multiple time-points.

For each of the above examples, it would be clearer if the in addition to the p values it could be indicated inside the brackets the statistical test used and the data groups being compared, or if multiple comparisons were performed and always produced the same result.

Reply: Thank you for this observation, and we also thought that it could be a good idea. However, the text was very confusing adding so many statistical data and p-values. To avoid this “messy”, the statistical tests were described below in the proper section “statistical analysis” and we adopted a common p-value among all of them (which could be p<0.05, but we preferred to be as closer as possible to the real p-value of each comparison). The entire Results section was rewritten to better describe what would be intra-group comparisons and between-group comparisons. It is important to cite that for the above-cited outcomes, which had non-normal distribution, it was used non-parametric tests: The Friedman test to verify differences among timepoints, and the Kruskal-Wallis test, which was used to compare groups. To be as clear as possible, we brought to the Results section only the p-values with statistically significant differences, avoiding reporting the outcomes attending the null hypothesis (which could be extremely confusing considering all 3 groups and 3 timepoints compared to baseline). We hope that the modifications meet your requirements.

Other comments

Page 2 Results Ln74 Briefly detail here the assignment of subjects to groups and define the abbreviations used for these groups. These are referred to throughout the Results but not described until the Materials and Methods, which is at the end of the report.

Reply: The assignment of subjects to groups and the definition of the abbreviations were added in the results section.

Page 3 Ln100 Fig 2 is now Fig 1.

Reply: This information was corrected.

Page 5 Ln 150 ‘its’ should be ‘is’ – ‘durability is of main importance’.

Reply: This information was corrected.

Page 5 Ln 152 delete comma ‘clinicians perceived’

Reply: This information was corrected.

Page 6 Ln196 after ‘tailored’ insert ‘to the severity of symptoms’

Reply: This information was corrected.

Page 6 Ln 215 change to ‘practitioners of routinary physical exercise’

Reply: This information was corrected.
